# Towards 4D Human Video Stylization

## Abstract

We present a first step towards 4D (3D and time) human video stylization, which addresses style transfer, novel view synthesis and human animation within a unified framework. While numerous video stylization methods have been developed, they are often restricted to rendering images in specific viewpoints of the input video, lacking the capability to generalize to novel views and novel poses in dynamic scenes. To overcome these limitations, we leverage Neural Radiance Fields (NeRFs) to represent videos, conducting stylization in the rendered feature space. Our innovative approach involves the simultaneous representation of both the human subject and the surrounding scene using two NeRFs. This dual representation facilitates the animation of human subjects across various poses and novel viewpoints. Specifically, we introduce a novel geometry-guided tri-plane representation, significantly enhancing feature representation robustness compared to direct tri-plane optimization. Following the video reconstruction, stylization is performed within the NeRFs' rendered feature space. Extensive experiments demonstrate that the proposed method strikes a superior balance between stylized textures and temporal coherence, surpassing existing approaches. Furthermore, our framework uniquely extends its capabilities to accommodate novel poses and viewpoints, making it a versatile tool for creative human video stylization. The source code and trained models will be made available to the public.

## 1 Introduction

Existing video style transfer methods have seen substantial progress in recent years (Li et al., 2019; Wang et al., 2020; Liu et al., 2021; Chen et al., 2021a; Chiang et al., 2022; Wu et al., 2022). These methods are designed to produce stylized frames given the content frames and a style image. To mitigate the issue of flickering artifacts between frames, they typically resort to optical flow or temporal constraints in an attempt to create smoother content. However, even as these methods excel at crafting high-quality stylized frames, built upon 2D networks, they are fundamentally bound to the same perspective as the source content videos. Consequently, video stylization in novel views remains unexplored. Furthermore, these techniques utterly lack the capability to alter or animate human poses within the stylized video, rendering them severely limited in their creative potential.

On the other hand, while NeRFs (Mildenhall et al., 2020) have been utilized in prior works to render stylized novel views in a *static scene* (Huang et al., 2022; Chiang et al., 2022; Liu et al., 2023) given dense views as the input, directly applying them to dynamic human scenes presents three primary issues. First, it is challenging to model *dynamic humans* in a scene across different frames, especially since NeRF is inherently designed for static scenes. Consequently, the learned model is incapable of performing human animation. Second, it is challenging to efficiently encode and optimize 3D points due to the significant computational cost associated with the model structure, such as multiplayer perceptions (MLPs). Third, one model for arbitrary style images (zero-shot stylization) presents an additional layer of complexity.

In this paper, we propose a holistic approach to perform video stylization on the original view, novel views, and animated humans by arbitrary styles through a united framework. Given a monocular video and an arbitrary style image, we first reconstruct both the human subject and environment simultaneously and then stylize the generated novel views and animated humans, facilitating creative effects and eliminating the dependency on costly multi-camera setups. More specifically, we incorporate a human body model, *e.g.*, SMPL (Loper et al., 2015), to transform the human from the video space to the canonical space, optimize the static human in the canonical space using NeRFs

| | Original Video | Novel View | Animation |
|---|:---:|:---:|:---:|
| LST | ✓ | ✗ | ✗ |
| AdaAttN | ✓ | ✗ | ✗ |
| CCPL | ✓ | ✗ | ✗ |
| Ours | ✓ | ✓ | ✓ |

(a) **Video stylization.** Showcasing the strengths in stylizing novel views and animations compared to 2D video stylization methods.

| | Novel View | Novel Pose | Dynamic scene |
|---|:---:|:---:|:---:|
| StylizedNeRF | ✓ | ✗ | ✗ |
| Style3D | ✓ | ✗ | ✗ |
| StyleRF | ✓ | ✗ | ✗ |
| Ours | ✓ | ✓ | ✓ |

(b) **NeRFs for stylization.** Demonstrating a unique capability in efficiently stylizing both novel views and poses within dynamic scenes.

Table 1: **Differences between our method and LST (Li et al., 2019), AdaAttN (Liu et al., 2021), CCPL (Wu et al., 2022), StylizedNeRF (Huang et al., 2022), Style3D (Chiang et al., 2022), and StyleRF (Liu et al., 2023).**

and make animation-driving of the human feasible. In addition, we improve the tri-plane-based representation (Fridovich-Keil et al., 2023) (representing the 3D space with three axis-aligned orthogonal planes which is fast in training and inference process)'s feature learning. We discretize both the human and scene spaces into 3D volumes and introduce the geometry prior by encoding the coordinates on the grids. This assists in learning a more robust feature representation across the entire 3D space. To render each pixel value, we project two camera rays into both NeRFs and extract the feature by projecting the points on each ray onto the three planes. Volume rendering is then utilized to generate a feature vector for each pixel, followed by the injection of VGG feature of the style image and the employment of a lightweight decoder to yield stylized RGB values. In summary, based on the NeRF models, our method can model both dynamic humans and static scenes in a unified framework, allowing high-quality stylized rendering of novel poses and novel views.

A naïve method to accomplish this new creative task might first utilize existing NeRF related method to generate the animated humans and novel views of a scene, followed by the application of existing video stylization techniques on these generated images. Our proposed method presents two notable advantages compared to it. First, by circumventing the use of predicted RGB images, our method applies stylization directly in the feature space, which tends to yield more consistent results compared to employing the generated images as input for video stylization. Second, our method enhances efficiency by eliminating the necessity for an image encoder, and instead, employs a lightweight decoder. This optimized architecture not only accelerates the processing speed but also maintains, if not enhances, the stylization quality across the diverse visual elements within the video narrative. The main contributions are outlined as follows:

- We propose a video stylization framework for dynamic scenes, which can stylize novel views and animated humans in novel poses, given a monocular video and arbitrary style images. While traditional video stylization methods are built upon 2D networks, ours is developed from a 3D perspective.
- We introduce a tri-plane-based representation and incorporate a geometric prior to model the 3D scenes. The representation is efficient and has better feature learning capability.
- Compared to existing methods, the proposed method showcases a superior balance between stylized textures and temporal coherence, and holds the unique advantage of being adaptable to novel poses and various backgrounds.

## 2 RELATED WORK

**Video stylization.** Video stylization extends image stylization (Gatys et al., 2016; Huang & Belongie, 2017) by enforcing temporal consistency of stylization across frames. This is achieved by exploring image-space constraints such as optical flow (Chen et al., 2017; Huang et al., 2017; Wang et al., 2020) and cross-frame feature correlation (Deng et al., 2021; Liu et al., 2021), as well as through careful design of feature transformations (Li et al., 2019; Liu et al., 2021). Nevertheless, stylization is confined to existing frames (or views) for the lack of a holistic scene representation. By contrast, our method presents the first 4D video stylization approach based on neural radiance fields. It is tailored for human-centric videos, achieves superior temporal consistency without image-space constraints, and uniquely supports stylized rendering of animated humans in novel views (Table 1a).

**Stylizing neural radiance fields.** Neural radiance fields (NeRFs) are volumetric scene representations introduced for novel view synthesis (Mildenhall et al., 2020). NeRF has lately been adapted

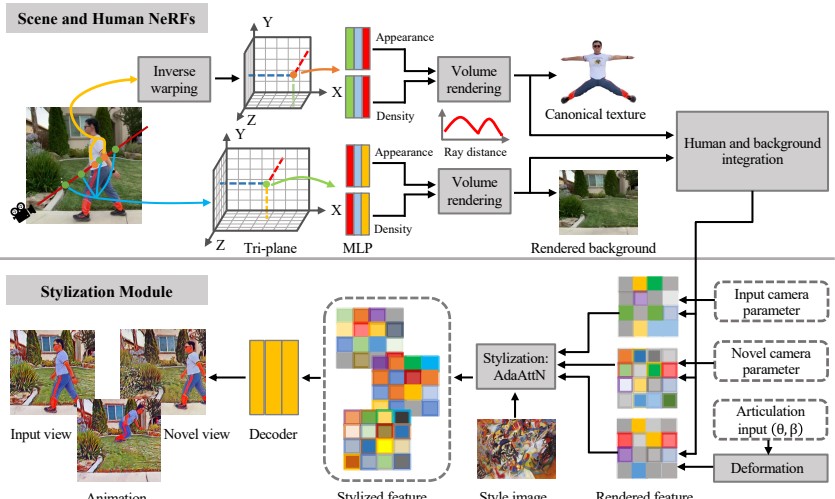

Figure 1: **Overview.** Given a camera ray, we sample foreground (human) and background (scene) points separately through the human and scene NeRFs. The points from the human are warped into canonical space via inverse warping. Then, each point is projected into three 2D planes to extract feature representation via bilinear interpolation, incorporated by Hadamard product. The features are utilized to predict the RGB appearance and density. We composite the foreground and background points for the dynamic foreground and multi-view background along each camera ray and apply volume rendering to attain the pixel feature on the 2D feature map. Subsequently, stylization is implemented on the feature map by AdaAttN (Liu et al., 2021), and a decoder is applied to process the stylized features, which are then decoded to the stylized image. Our model can stylize novel views and animated humans in the same scene by giving the novel camera parameters and articulation as extra inputs.

for stylized novel view synthesis (Chiang et al., 2022; Zhang et al., 2022; Xu et al., 2023; Liu et al., 2023) as it provides strong geometric constraints to enforce multi-view consistency of stylization. Early methods bake the style into the weights of a NeRF and thus require learning one model for each style (Zhang et al., 2022; Xu et al., 2023). Most relevant to our work, StyleRF (Liu et al., 2023) enables zero-shot NeRF stylization via deferred style transformation, where 2D feature maps volume-rendered from a NeRF are modulated by an arbitrary style and subsequently decoded into a stylized image. Similar to StyleRF, our method leverages NeRF as the underlying scene representation and supports zero-shot stylization. Different from StyleRF, our method takes as input a monocular video of a moving human as opposed to multi-view images of a static scene (Table 1b).

## 3 METHODOLOGY

Given a monocular video of a dynamic human and an arbitrary style image, our goal is to synthesize *stylized novel views* of a person with any *different poses* (i.e., animated humans) in a scene. To achieve this, we propose a unified framework (Figure. 1) consisting of three modules: 1) two novel tri-plane based feature representation networks to encode geometric and appearance information of dynamic humans and their surroundings; 2) a style transfer module to modulate the rendered feature maps from NeRFs as conditioned on the input style image, 3) a lightweight decoder to synthesize the stylized images from novel viewpoints or new poses. We will present each module in detail.

### 3.1 GEOMETRY GUIDED TRI-PLANE BASED FEATURE REPRESENTATION

Motivated by (Fridovich-Keil et al., 2023), we incorporate a tri-plane representation to model the 3D scene in canonical space, thereby reducing memory usage and accelerating the training and rendering processes compared to MLPs utilized in NeRF. The tri-plane representation describes a scene with three orthogonal planes ($\mathbf{P}_{xy}, \mathbf{P}_{xz}, \mathbf{P}_{yz}$). For any 3D point, we project its 3D coordinate onto the three orthogonal planes to get corresponding locations in each plane. Then, the features of the 3D point are computed as the product of bilinearly interpolated features on three planes. A small MLP is used to decode the features into density and appearance.

However, tri-plane features without spatial constraints are limited in expressiveness when directly optimized. This can be verified from Figure 2 that NeRF with tri-plane produces blurry results on

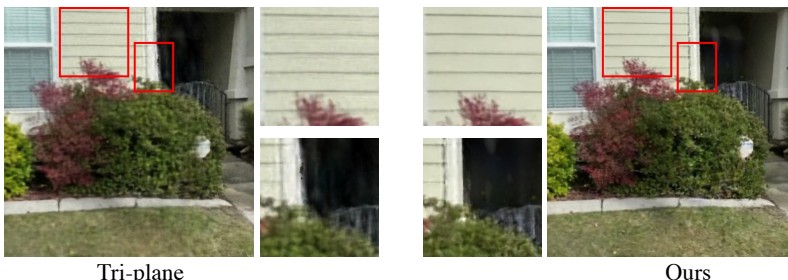

Tri-plane                Ours

Figure 2: **Visual results of direct optimization using the tri-plane features and the geometry guided tri-plane features.** Our method can recover more clear background texture (1st row) and sharper contours (2nd row).

the wall and its boundary. To overcome this limitation, we propose to encode the 3D coordinates anchored to the tri-plane as the geometric prior over the whole space. Here we discretize the 3D space as a volume and divide it into small voxels, with sizes of 10 mm $\times$ 10 mm $\times$ 10 mm. Voxel coordinates transformed by the positional encoding $\gamma_v(\cdot)$ are mapped onto three planes to serve as the input to the tri-plane. Encoded coordinates projected onto the same pixel are aggregated via average pooling, resulting in planar features with size $H_{\mathbf{P}_i} \times W_{\mathbf{P}_i} \times D$, where $\mathbf{P}_i$ represents the $i$-th plane and $D$ is the dimension of the feature on each plane. Motivated by U-Net architecture (Ronneberger et al., 2015), we use three encoders with 2D convolutional networks to represent the tri-plane features.

To obtain the feature $f_p(\boldsymbol{x})$ of a 3D point $\boldsymbol{x} = (x, y, z)$, we project the point onto three planes $\pi_p(\boldsymbol{x})$, $p \in (\mathbf{P}_{xy}, \mathbf{P}_{xz}, \mathbf{P}_{yz})$, where $\pi_p$ presents the projection operation that maps $\boldsymbol{x}$ onto the $p$'th plane. Then, bilinear interpolation of a point is executed on a regularly spaced 2D grid to obtain the feature vector by $\phi(\pi_p(\boldsymbol{x}))$. The operation of each plane is repeated to obtain three feature vectors $f_p(\boldsymbol{x})$. To incorporate these features over three planes, we use the Hadamard product (element-wise multiplication) to produce an integrated feature vector, $f(\boldsymbol{x}) = \prod_{p \in (\mathbf{P}_{xy}, \mathbf{P}_{xz}, \mathbf{P}_{yz})} f_p(\boldsymbol{x})$.

Finally, $f(\boldsymbol{x})$ will be decoded into color and density using two separate MLPs. Either Hadamard product or additional operations can be utilized to generate the feature vector $f(\boldsymbol{x})$. We choose the Hadamard product here as it can generate spatially localized signals, which is a distinct advantage over addition, as described in (Fridovich-Keil et al., 2023). Figure 2 shows the proposed geometry guided method generates clearer background pixels than those by the one with direct optimization. Quantitative results on stylization can be found in Section 4.3.

## 3.2 NEURAL RADIANCE FIELDS

We propose to leverage the NeRF to model 3D scenes and extend the original NeRF (Mildenhall et al., 2020) to achieve the purpose of stylization. In the original NeRF, the color and density are the direct output for any queried 3D point. But for each point on the camera ray, we predict a $C$-dimensional feature vector $\hat{f}(\boldsymbol{x}) \in \mathbb{R}^C$, motivated by (Niemeyer & Geiger, 2021; Liu et al., 2023). Specifically, for every queried point $\boldsymbol{x} \in \mathbb{R}^3$, our model outputs its volume density $\sigma$ and feature vector $\hat{f}(\boldsymbol{x})$ by $F_\Theta : (f(\boldsymbol{x}), \gamma_d(\boldsymbol{d})) \to (\sigma, \hat{f}(\boldsymbol{x}))$, where $\gamma_d(\boldsymbol{d})$ represents the positional encoding on the view direction $\boldsymbol{d}$ and $f(\boldsymbol{x})$ is the feature vector extracted from the tri-plane. Then, the feature vector of any image pixel is derived by accumulating all $N$ sampled points along the ray $\boldsymbol{r}$ through integration (Mildenhall et al., 2020),

$$f(\boldsymbol{r}) = \sum_{i=1}^{N} w_i \hat{f}(\boldsymbol{x_i}), \quad w_i = T_i \left(1 - \exp\left(-\sigma_i \delta_i\right)\right), \tag{1}$$

where $\sigma_j$ and $\delta_i$ denote the volume density and distance between adjacent samples, $w_i$ is the weight of the feature vector $\hat{f}(\boldsymbol{x_i})$ on the ray $\boldsymbol{r}$, and $T_i = \exp(-\sum_{j=1}^{i-1} \sigma_i \delta_i)$ is the accumulated transmittance along the ray. We treat the background across frames as multi-view images and train the scene NeRF for the background model with the human masked out in each frame. To capture dynamic humans with various poses, we train the human NeRF in the canonical space, leveraging priors with the deformation field by transforming the human from observation to canonical space. Here, the observation space describes the images from the input video, and the canonical space is the global one shared by all frames.

**Scene NeRF.** We extract features from the background tri-plane to predict the density and feature vector $\hat{f}_b(\boldsymbol{x})$ for each point with two tiny MLP networks. In detail, the density branch has one fully connected layer, while the feature branch utilizes a hidden layer comprising 128 units, followed by one output layer. Subsequently, the feature vectors on the same camera ray are aggregated to generate a feature vector $f_b(\boldsymbol{r})$ for each pixel using Equation 1.

**Human NeRF.** The human NeRF is represented in a 3D canonical space. To synthesize a pixel on the human body for each video frame as in the observation space, the sampled points along the corresponding ray are transformed from the observation space into the canonical space by the rigid transformation associated with the closest point on the mesh. Here, we use a parametric SMPL (Loper et al., 2015) model to provide explicit guidance on the deformation of spatial points. This approach is beneficial for learning a meaningful canonical space while simultaneously reducing dependency on diverse poses when generalized to unseen poses. This allows us to train the NeRF in dynamic scenarios featuring a moving person and animate the person during inference.

Motivated by (Chen et al., 2021b), the template pose in canonical space is defined as X-pose $\theta_c$ because of its good visibility and separability of each body component in the canonical space. The pose $\theta_o$ in the observation space can be converted to X-pose $\theta_c$ in the canonical space by using the inversion of the linear skinning derived from the SMPL model. We extend these transformation functions to the space surrounding the mesh surface to allow the 3D points near the mesh to consistently move with adjacent vertices.

Specifically, the inverse linear blend skinning is defined based on the 3D human skeleton. The human skeleton represents $K$ parts that generate $K$ transformation matrices $\{G_k\} \in SE(3)$: $\tilde{\boldsymbol{x}} = \left(\sum_{k=1}^{K} w_o(\boldsymbol{x})_k G_k\right)^{-1} \boldsymbol{x}$, where $w_o(\boldsymbol{x})_k$ is the blend weight of the $k$-th part. $\boldsymbol{x}$ and $\tilde{\boldsymbol{x}}$ denote the 3D points in observation and canonical spaces respectively. This inverse function cannot fully express the deformation details caused by the rather complicated movement of clothes and misaligned SMPL poses. Thus, motivated by (Jiang et al., 2022), we adopt an error-correction network $\mathcal{E}$ to correct the errors in the warping field, which learns a mapping for a point from the observation space to the canonical space. Here, $\mathcal{E}$ comprises an MLP with the 3D point coordinates as the input and predicts the error. Therefore, the canonical point will be defined as $\tilde{\boldsymbol{x}}_c = \tilde{\boldsymbol{x}} + \mathcal{E}(\boldsymbol{x})$. For each point $\tilde{\boldsymbol{x}}_c$, we extract features from tri-plane of the human and utilize another lightweight decoder to predict the density and appearance $\hat{f}_h(\boldsymbol{x})$.

**Composite NeRFs.** To obtain the RGB value for each image pixel, two rays, one for the human NeRF and the other for the scene NeRF, are utilized. Colors and densities for the points on the two rays are obtained and are sorted in an ascending order based on the depth values. We then utilize Equation 1 to render a feature vector $f(\boldsymbol{r})$ based on the points on the two rays.

### 3.3 STYLIZING THE SCENE

For the stylization module, it takes the above-mentioned NeRF rendered features (content features) and the style image as input and generates the stylized image based on the Adaptive Attention Normalization (AdaAttN) layer (Liu et al., 2021). The feature map of the style image is extracted from the pre-trained VGG network (Simonyan & Zisserman, 2014). Let $F_s$ be the style features and $F_c$ be the set of content features tailored to a 2D patch. Each pixel vector is rendered by **Composite NeRFs** introduced in Section 3.2. The style transfer module is formulated by $F_{cs} = \psi(\text{AdaAttN}(\phi(F_c), F_s))$, where $\phi$ and $\psi$, formulated as MLPs, are learned mappings for the content and stylized features, and AdaAttN (Liu et al., 2021) is designed to adaptively transfer the feature distribution from the style image to the content by the attention mechanism. Specifically, $F_{cs}$ is generated by calculating the attention map within the features of the content and style images. The stylized feature $F_{cs}$ will then be applied to generate the stylized image via a decoder.

### 3.4 IMAGE DECODING

Finally, an image decoder $F_\theta$ is designed to mapping the stylized 2D feature $F_{cs} \in \mathbb{R}^{H \times W \times M}$ that captures high-level information to the final stylized image $I \in \mathbb{R}^{H \times W \times 3}$ at input resolution,

$$F_\theta : \mathbb{R}^{H \times W \times M} \to \mathbb{R}^{H \times W \times 3}. \tag{2}$$

The operation $F_\theta$ comprised of convolutional and ReLU activation layers, aiming to render a full-resolution RGB image, is parameterized as a 2D decoder. In the convolutional layer, we opt for $3 \times 3$ kernel sizes without intermediate layers to only allow for spatially minor refinements to avoid entangling global scene properties during image synthesis.

### 3.5 OBJECTIVE FUNCTIONS

We aim to stylize novel views of animated humans based on the reconstructed scene and humans. In the reconstruction stage, we first train the NeRFs by minimizing the rendered RGB image reconstruction loss. Afterward, in the stylization stage, we remove the last fully connected layer of the above-trained NeRF networks and attach the stylization module and the decoder to synthesize stylized images. Next, we introduce the losses adopted for training both the scene and human NeRFs and the objective functions for stylization.

**Scene NeRF.** The objective function for training the scene NeRF is defined as

$$\mathcal{L}_s(\boldsymbol{r}) = \sum_{\boldsymbol{r} \in \mathcal{R}} ||C_s(\boldsymbol{r}) - \tilde{C}_s(\boldsymbol{r})||, \tag{3}$$

where $R$ is the set of rays. $C_s$ and $\tilde{C}_s$ denote the prediction and the ground truth RGB values, respectively.

**Human NeRF.** The region covered by the human mask $\mathcal{M}(\cdot)$ is optimized by

$$\mathcal{L}_r(\boldsymbol{r}) = \mathcal{M}(\boldsymbol{r})||C_h(\boldsymbol{r}) - \tilde{C}_h(\boldsymbol{r})||, \tag{4}$$

where $C_h(\boldsymbol{r})$ and $\tilde{C}_h(\boldsymbol{r})$ are the rendered and ground truth RGB values. More losses to train the Human NeRF can be found in the appendix.

After training the scene and human NeRFs for reconstruction, we discard the last fully connected layer. Then, the feature vector and density for sampled points are aggregated to render the feature vector as introduced in **Composite NeRFs** of Section 3.2. Finally, a decoder with convolutional and nonlinear layers converts the feature map into an RGB image. Here, two losses are applied, aiming to reconstruct the input video and also acquire semantic information for the feature patch $F_c$,

$$\mathcal{L}_v = ||F_c - \tilde{F}_c|| + \sum_{l \in l_p} ||F^l(I) - F^l(\tilde{I})|| + ||I - \tilde{I}||, \tag{5}$$

where $F_c$ and $\tilde{F}_c$ denote the rendered feature map and the feature extracted from the pretrained VGG network. $F(I)$ and $F(\tilde{I})$ are the predicted features and the features extracted from VGG with the RGB image as the input, respectively. In addition, $I$ and $\tilde{I}$ are the predicted and ground truth images. $l_p$ denotes the set of VGG layers.

**Stylization.** We use the content and style losses from AdaAttN (Liu et al., 2021), encompassing both global style and local feature losses. The former ensures a global stylized effect, and the latter can generate better stylized output for local areas.

## 4 EXPERIMENTS

**Implementation.** We train our model in two stages: video reconstruction and stylization. In the *reconstruction* stage, the model is trained to predict input video frames. This facilitates the synthesis of novel views and further enables human animation. We apply the losses in Equation 3 and Equation 4 to minimize view reconstruction error and learn human pose transformations between the observation and canonical spaces. Once the training of scene and human NeRFs converges, we render the feature map of a sampled patch, which serves as the input to a 2D decoder that predicts the RGB values. Here, we freeze the density branch and the layers shared by the density and appearance branches. The subsequent layers are optimized using the losses in Equation 5. For the *stylization* stage, we utilize the content and style losses in AdaAttN as introduced in Section 3.5.

We obtain a set of $N$ points using stratified sampling for both scene and human NeRFs, where $N$ is set to 128. All layers are trained using Adam (Kingma & Ba, 2015). The learning rate for frame reconstruction starts at $1 \times 10^{-4}$ and decays exponentially over the process of training. The learning rate for the stylization stage is set to $2 \times 10^{-5}$.

**Run time.** Compared to the NeRF approaches that use MLPs to learn the feature representation for each sample point along the camera ray, the proposed tri-plane based representation significantly accelerates rendering, achieving a speedup of approximately 70% at inference time.

**Datasets.** We utilize two datasets of monocular videos, including NeuMan (Jiang et al., 2022) and a dataset captured by our smartphone. The first dataset comprises six videos with 40 to 104 frames. It includes indoor and outdoor scenes with diverse human subjects of various genders and races. However, this dataset presents two primary limitations. First, frames extracted from longer videos produce less fluid transitions across frames. Second, the limited number of frames makes it difficult to evaluate the robustness of our proposed method. To compensate for this, we capture two additional videos, emphasizing longer duration and more diverse scenes with about 150-200 frames.

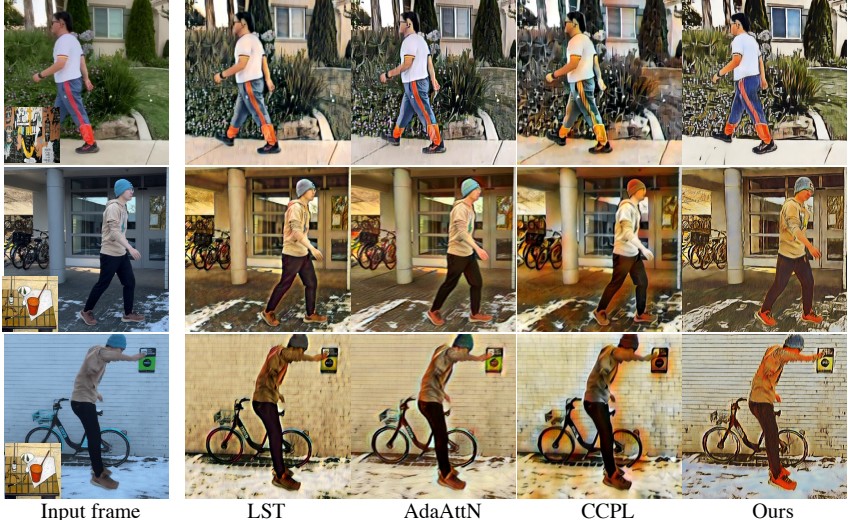

Figure 3: **Visual comparison with state-of-the-art methods.** The first two rows show the stylization results given the captured video frame as the input for 2D video stylization methods. The last row utilizes the animated human generated by our method as the input. All results demonstrate the efficacy of the proposed method in generating the patterns and textures of the style images.

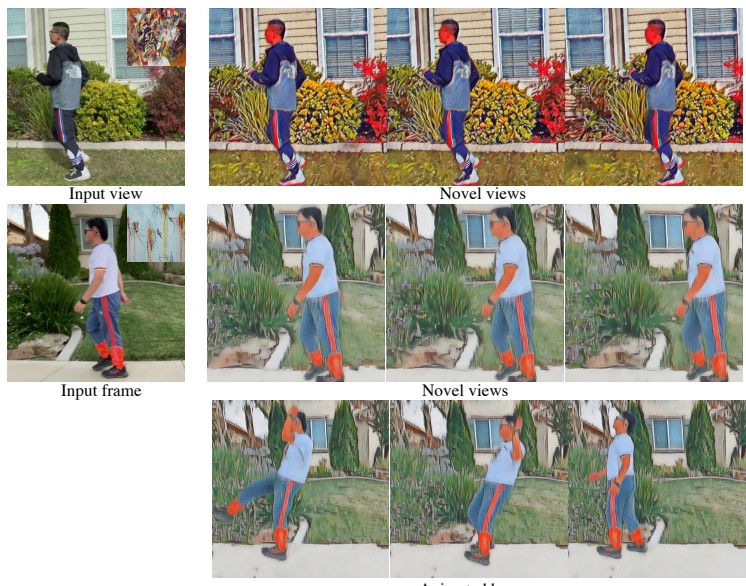

Figure 4: **Examples about the novel view synthesis and animation.** The first two rows shows the novel view results around the human by moving the camera from right to left. The third row visualizes the stylized human given different poses.

## 4.1 QUALITATIVE RESULTS

**Comparison with state-of-the-art stylization methods.** We present visual comparison with state-of-the-art 2D video stylization methods LST (Li et al., 2019), AdaAttN (Liu et al., 2021), and CCPL (Wu et al., 2022) in Figure 3. It can be seen that the proposed method achieves better stylization on different subjects, as depicted in the 1st and 2nd rows. Our method can produce stylization with textures and patterns much more similar to the style images. In contrast, LST (Li et al., 2019) and AdaAttN (Liu et al., 2021) transfer fewer textures from the style image. Both LST (Li et al., 2019) and CCPL (Wu et al., 2022) generate blurry stylized images and exhibit more artifacts, particularly on the human and ground as seen in the 1st row.

**Novel view synthesis and animation.** Unlike existing video stylization methods that perform stylization on the input view, our model is capable of stylizing images from novel views and novel

Table 2: **Temporal consistency with 2D video stylization methods.** Consistency is calculated by warping error (↓). The best and second best performances are in red and blue colors.

| Models | LST [CVPR2019] | AdaAttN [ICCV2021] | CCPL [ECCV2022] | **Ours** |
|--------|----------------|--------------------|-----------------|----------|
| *Our dataset* | 0.169 | 0.161 | 0.231 | 0.165 |
| *NeuMan* | 0.226 | 0.239 | 0.298 | 0.214 |

Table 3: **Temporal consistency with 2D video stylization methods on novel views and animated humans.** The input of all methods are generated by the proposed method. Consistency is calculated by warping error (↓). The best and second best performances are in red and blue colors.

| LST[CVPR2019] | AdaAttN[ICCV2021] | IEContraAST[NeurIPS2021] | CSBNet[IJCAI2022] | CCPL[ECCV2022] | **Ours** |
|---------------|-------------------|--------------------------|-------------------|----------------|----------|
| 0.159 | 0.152 | 0.269 | 0.247 | 0.239 | 0.143 |

Table 4: **Quantitative results on temporal consistency (lower is better).** The proposed method designed for dynamic scenes achieves much better performance compared to StyleRF.

| | *Our dataset* | *NeuMan* |
|--------------|---------------|-----------------|
| StyleRF / Ours | 0.293 / 0.165 | 0.387 / 0.214 |

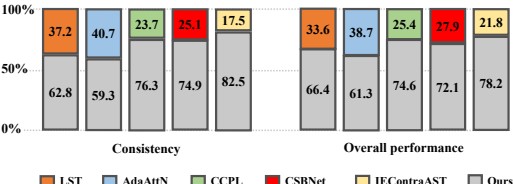

Figure 5: **User study on the original video space.** We present videos produced by two methods each time, and ask each volunteer to select the one with less flickering (consistency) and a better balance of temporal coherence and style quality (overall performance).

Figure 6: **User study on novel views and animated humans.** We present videos produced by two methods each time, and ask each volunteer to select the one with less flickering (consistency) and a better balance of temporal coherence and style quality (overall performance). The inputs of all methods are predicted by the proposed method.

poses, which benefits from the utilization of human and scene NeRFs. Once our model is adequately trained, it can seamlessly synthesize novel views and animate humans during inference. Visual examples can be found in Figure 4.

## 4.2 QUANTITATIVE RESULTS

**Consistency evaluation.** To quantify the consistency across frames, following (Liu et al., 2023), we leverage the optical flow, warp one frame to the subsequent one, and then compute the masked LPIPS score (Zhang et al., 2018). The consistency scores are obtained by comparing adjacent views and far-away views, respectively. The average results of these comparisons are presented in Table 2, Table 3 and Table 4. We compare the proposed method against state-of-the-art video stylization methods including LST (Li et al., 2019), AdaAttN (Liu et al., 2021), IEContraAST (Chen et al., 2021a), CSBNet (Lu & Wang, 2022) , CCPL (Wu et al., 2022) and one NeRF-based multi-view method StyleRF (Liu et al., 2023). Compared to the 2D video stylization methods, our method shows better performance for consistency, which benefits from the consistent geometry learned by NeRF. The proposed method designed for dynamic scenes achieves much better performance compared to StyleRF.

**User study.** We conduct a user study to gain deeper insights into the perceptual quality of the stylized images produced by our method in comparison to the baseline methods. Our study is organized into two sections: temporal consistency and overall synthesis quality. We visualize the results in Figure 5 and Figure 6 with approximately 3000 votes and 5000 votes, respectively. Figure 5 shows

Table 5: **Ablation study with video stylization methods on temporal consistency.** By replacing the input of the 2D stylization methods with the rendered images by our method, we demonstrate that our unified framework can generate better results compared to the combination of NeRFs and 2D stylization methods. The best and second best performance are in red and blue colors.

| Models | LST [CVPR2019] | AdaAttN [ICCV2021] | CCPL [ECCV2022] | **Ours** |
|---|---|---|---|---|
| *Our dataset* | 0.185 | 0.179 | 0.261 | 0.165 |
| *NeuMan* | 0.248 | 0.267 | 0.321 | 0.214 |

Table 6: **Ablation study with the vanilla tri-plane (Fridovich-Keil et al., 2023) on temporal consistency (lower is better).** The proposed geometry-guided tri-plane encoded by U-Nets achieves better consistency than directly optimizing the features of the tri-plane (Fridovich-Keil et al., 2023).

| Tri-plane (Fridovich-Keil et al., 2023) | **Ours** |
|---|---|
| 0.207 | **0.182** |

results on the original 2D videos and Figure 6 shows results on novel views and animated humans generated by the proposed method. On the original video space (Figure 5), our method outperforms the baseline methods in terms of overall synthesis quality, which underscores the efficacy of our method in striking a delicate balance between temporal coherence and stylization quality. On the novel views and animated humans (Figure 6), our method shows superior performance on all metrics, which demonstrates the efficacy of the proposed unified framework. More details can be found in supplementary material.

### 4.3 ABLATION STUDIES

In this work, we propose to addresses style transfer, novel view synthesis and human animation within a unified framework. An alternative approach could employ an existing dynamic NeRF-based method to predict animated humans and novel views in a scene, which is then followed by applying existing 2D video stylization methods. Here, we render frames with animated humans utilizing our first-stage method and then apply the rendered frames as the input to video stylization methods. The visual comparison is illustrated in the last row of Figure 3. As observed, our method paints the scene in the desired style while preserves the structure of the content image. Quantitative results can be found in Table 5, which demonstrates the advantages of our proposed unified framework.

In addition, to demonstrate the efficacy of the proposed geometry-guided tri-plane, we show quantitative results in Table 6. It can be seen that the proposed geometry-guided tri-plane can generate better consistency than the vanilla tri-plane (Fridovich-Keil et al., 2023), which demonstrates the advantages of U-Nets to encode the tri-plane features than directly optimizing tri-plane features in (Fridovich-Keil et al., 2023). Visual results can be found in supplementary material.

### 5 CONCLUSION

Our work looks into the problem of video stylization, with particular emphasis on dynamic humans. Going beyond the existing video stylization, we have proposed a unified framework for 3D consistent video stylization which also supports flexible manipulation of viewpoints and body poses of humans. To accomplish this, we incorporate NeRF representation to encode both the human subject and its surroundings and conduct stylization on the rendered features from NeRF. Specifically, a geometry-guided tri-plane representation is introduced to learn the 3D scene in a more efficient and effective manner. We have demonstrated both our superior performance on stylized textures and long-term 3D consistency with the unique capability of conducting novel view and animated stylization with extensive evaluations.

**Limitations and future directions.** First, our current approach is constrained by the limited variation in camera pose and human face angles within the input video, restricting novel views to smaller angles. Future research can explore generative techniques to extrapolate unseen backgrounds and human features, enabling the creation of more expansive novel views. Second, while our current implementation has been optimized for speed, it still falls short of supporting real-time manipulation. One potential avenue for improvement is to pre-render stylized features and then reuse them across different views and various human poses to enhance real-time performance. Third, our method achieves the best trade-off between stylization and consistency. A future research direction could focus on achieving the utmost stylization effect without compromising consistency or style quality.

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

## A  APPENDIX

### A.1  LOSSES FUNCTIONS

We train the proposed method using two stages including the input reconstruction and feature stylization.

To reconstruct the input, we utilize the vanilla NeRFs with MLPs to predict the RGB value for the human and scene. Next we will introduce the losses that are utilized for training the NeRFs.

**Scene NeRF.** The objective function for training the scene NeRF is defined as

$$\mathcal{L}_s(\boldsymbol{r}) = \sum_{\boldsymbol{r}\in\mathcal{R}} ||C_s(\boldsymbol{r}) - \tilde{C}_s(\boldsymbol{r})||, \tag{6}$$

where $R$ is the set of rays. $C_s$ and $\tilde{C}_s$ denote the prediction and the ground truth RGB values.

**Human NeRF.** The region covered by the human mask $\mathcal{M}(\cdot)$ is optmized by

$$\mathcal{L}_r(\boldsymbol{r}) = \mathcal{M}(\boldsymbol{r})||C_h(\boldsymbol{r}) - \tilde{C}_h(\boldsymbol{r})||, \tag{7}$$

where $C_h(\boldsymbol{r})$ and $\tilde{C}_h(\boldsymbol{r})$ are the rendered and ground truth RGB values.

We use $\mathcal{L}_a$ to enforce the accumulated alpha map from the human NeRF to be similar to the detected human mask,

$$\mathcal{L}_a(\boldsymbol{r}) = \mathcal{M}(\boldsymbol{r})||1 - \alpha_h(\boldsymbol{r})||, \tag{8}$$

where $\alpha_h(\cdot)$ corresponds to accumulated density over the ray.

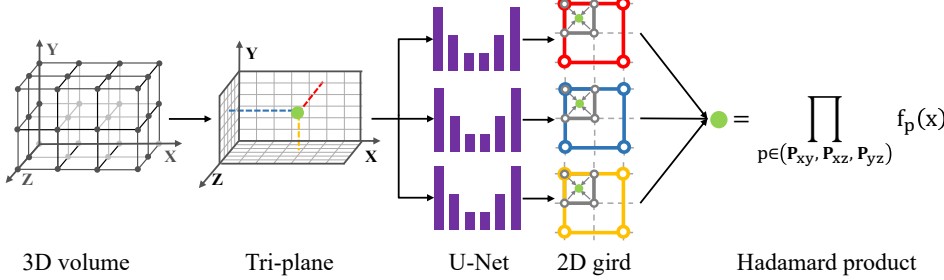

Figure 7: **Feature extraction from the tri-plane.**

To remove artifacts such as blobs in the canonical space and semi-transparent canonical humans, we enforce the volume $\sigma_h$ inside the canonical SMPL mesh to be solid while enforcing the volume $\sigma_h$ outside the canonical SMPL mesh to be empty:

$$\mathcal{L}_{smpl}(\tilde{\boldsymbol{x}}, \sigma_h) = \begin{cases} \|1 - \sigma_h\|, & \text{if } \tilde{\boldsymbol{x}} \text{ inside SMPL mesh} \\ |\sigma_h|, & \text{otherwise.} \end{cases} \tag{9}$$

This loss function encourages the proposed method to model shapes as solid surfaces (i.e., sharp outside-to-inside transitions).

To encourage the solid surfaces in the canonical human, we encourage the weight of each sample to be either 1 or 0, given by

$$\mathcal{L}_{hard} = -\log(e^{-|w|} + e^{-|1-w|}), \tag{10}$$

where $w$ is derived from an accumulation of the transmittance along the camera ray as defined in Equation 1. In addition, a canonical edge loss $\mathcal{L}_{edge}$ is utilized to predict a sharp canonical shape. This is carried out by raycasting a random line in the canonical volume and encouraging the alpha $\alpha_c$ values to be 1 or 0:

$$\mathcal{L}_{edge} = -\log(e^{-|\alpha_c|} + e^{-|1-\alpha_c|}). \tag{11}$$

Overall, the losses for our model are formulated by

$$\mathcal{L}_h = \mathcal{L}_r + \lambda_a \mathcal{L}_a + \lambda_{smpl} \mathcal{L}_{smpl} + \lambda_{hard} \mathcal{L}_{hard} + \lambda_{edge} \mathcal{L}_{edge}. \tag{12}$$

we set $\lambda_{mask} = 0.01$, $\lambda_{smpl} = 1.0$, $\lambda_{hard} = 0.1$, and $\lambda_{edge} = 0.1$. The $\lambda_{mask}$ linearly decays to 0 throughout the training since the detected masks are inaccurate.

## A.2 USER STUDY

We utilize 32 sequences across two datasets and invited participants for evaluation. Initially, we presented participants with an input video and a style image, alongside two stylized videos—one generated by our method and the other is a randomly selected comparison method. Subsequently, we asked the participants to select the options based on two evaluative indicators: temporal consistency, and overall performance. We collect about 3000 votes and summarize the results as shown in the main paper.

## A.3 FEATURE EXTRACTION FROM THE TRI-PLANE.

In this paper, we propose to learn a more robust feature representation by the geometric prior on the tri-plane. As shown in Figure 7, we discretize the 3D space as a volume and divide it into small voxels, with sizes of 10 mm × 10 mm × 10 mm. Voxel coordinates transformed by the positional encoding $\gamma_v(\cdot)$ are mapped onto three planes to serve as the input to the tri-plane. Encoded coordinates projected onto the same pixel are aggregated via average pooling, resulting in planar features with size $H_{\mathbf{P}_i} \times W_{\mathbf{P}_i} \times D$, where $D$ is the dimension of the feature on each plane. Motivated by U-Net architecture (Ronneberger et al., 2015), we use three encoders with 2D convolutional networks to represent the tri-plane features.

To obtain the feature $f_p(\boldsymbol{x})$ of a 3D point $\boldsymbol{x} = (x, y, z)$, we project the point onto three planes $\pi_p(\boldsymbol{x})$, $p \in (\mathbf{P}_{xy}, \mathbf{P}_{xz}, \mathbf{P}_{yz})$, where $\pi_p$ presents the projection operation that maps $\boldsymbol{x}$ onto the $p$'th

plane. Then, bilinear interpolation of a point is executed on a regularly spaced 2D grid to obtain the feature vector by $\phi(\pi_p(\boldsymbol{x}))$. The operation of each plane is repeated to obtain three feature vectors $f_p(\boldsymbol{x})$. To incorporate these features over three planes, we use the Hadamard product (element-wise multiplication) to produce an integrated feature vector,

$$f(\boldsymbol{x}) = \prod_{p \in (\mathbf{P}_{xy}, \mathbf{P}_{xz}, \mathbf{P}_{yz})} f_p(\boldsymbol{x}). \tag{13}$$

Next, $f(\boldsymbol{x})$ will be decoded into color and density using two separate MLPs.

