# OpenReview forum: "Towards 4D Human Video Stylization"
_ICLR.cc/2024/Conference — Submitted to ICLR 2024_

### Official Review · Reviewer_smP6 · 2023-10-25

**Soundness:** 3 good
**Presentation:** 3 good
**Contribution:** 3 good
**Rating:** 5
**Confidence:** 3

**Summary:**

This paper desribes a NeRF based video stylization methods. The authors propose to split human action and background scenes into two different NeRF to synthesis novel views and a novel geometry-guided tri-plane representation to enhanced feature representation robustness and synthesis quality. The stylization is then performed within the NeRFs’ rendered feature space. Both qualitative and quantitative experiments have been conducted to demonstrate that the proposed method outperforms existing approaches.

**Strengths:**

1.By spliting human gesture and background scene into two different NeRF branches, the proposed method can effective synthesis novel views and animated human.
2.The proposed geometry-guided tri-plane representation can effectively stablize image synthesis and improve result quality.
3.Compared to previous methods, the generated samples have better visual quality

**Weaknesses:**

1.While there are plenty of video stylizatio works, in the experiment section, the authors only compare the proposed method with 3 exsisting methods, which seems to be not adequate.

2.In both qualitative comparison (table2) and user study (Figure 5), the advantage of proposed method over previous seems not very obvious. Especially in user study, the proposed method only have obvious advantage over CCPL but received similar evaluation compared to LST and AdaAttN

3.The authors are putting "4D" in the title, trying to emphesize the contribution on video stylizaiton. Personally I don't think it's proper to regard the video stylization as a main contribution of this paper, as the proposed method directly apply stylization on the NeRF rendered feature space and doesn't have any mechanism for inter-frame stablizaiton. Also, the novel view synthesis / human animation function and stylization seems to have little connection.

**Questions:**

The stylization is directly applied on the NeRF rendered feature space, and seems to have littile connection to the proposed geometry-guided tri-plane representation and two branch NeRF for human and background scene. Also, in the METHODOLOGY section, the authors are repeatedly mentioning that many components of the framework are motived by previous methods. How could authors persuade reviewers the proposed the framework is innocative instead of an incremental work with the combination of previous works?

---

> ### Author Response · Authors · 2023-11-23
>
> **[Q1] Comparison with more methods.**
>
> We present 2 more methods in Table 3 & Figure 6. We use 50 videos on two datasets and collect about 5000 votes for user study. Visual comparison with state-of-the-art is shown in the video [01:20-03:29].
>
>
> **[Q2] In Table 2 & Figure 5, the advantage seems not very obvious.**
>
> **The advantage of our method is not obvious compared with certain methods because** the inputs of the proposed method and others are different. 2D video stylization methods use the original video as the input. Our method uses NeRFs to reconstruct the input and then stylizes the reconstructed video. The reconstruction would bring inconsistency because training frames are very sparse. However, despite this limitation, the proposed method still performs better than state-of-the-art.
>
> **The advantage of our method is quite obvious when all methods use the same input**, which can be verified in the ablation studies (Table 5). By replacing the input of the 2D stylization methods with the **reconstructed videos by our method**, our unified framework can generate more consistent results. To further show our superior performance on novel views and animated humans, we render another three videos and show the results in Table 3 and Figure 6. Here all methods use the same inputs, which are predicted by the proposed method. We can see that the proposed method strikes a superior balance between stylized textures and temporal coherence. Visual examples can be found in the supplementary video [01:20-03:29].
>
> **[Q3] "4D" in the title and inter-frame stabilization. Connection between the stylization in the rendered feature map, synthesis/human animation function and the proposed geometry-guided tri-plane representation.**
>
> We adopt "4D" to highlight our contribution: the first method capable of stylizing entire scenes (human & background), under novel views and poses aside from the original videos. This has not yet explored in existing methods.
>
> The inter-frame stabilization is ensured by two-step training. We first train NeRFs without stylization and decoder modules to reconstruct the input. Then we fix the density branch to optimize the stylization and decoder modules. Experiments in Table 2 & 3 show that our method generates better consistency than state-of-the-art.
>
> **The connection between stylization in the feature map and novel view synthesis/animation is built on efficacy via a unified framework.** Typically, novel view synthesis/human animation and stylization can be performed separately—first by employing NeRFs to predict novel views or animate humans, and then by applying 2D video stylization methods. However, this separate approach tends to result in inconsistency compared to our unified framework. This is verified by ablation studies in Table 5.
>
> **The connection between stylization in the feature map and tri-plane representation is built on the same goal-efficiency.** There are two ways to achieve stylization with NeRFs: stylization on the 3D points or on the rendered map. We opt for the second way for the first one is computationally intensive. In addition, tri-plane is utilized for efficiency compared to MLPs when numerous points are sampled. Both stylization strategy and tri-plane representation contributes to efficient learning.
>
>
> **[Q4] Novelty.**
>
> While some components are known in the field, it requires **meticulous algorithmic design** to develop an efficient and effective model to synthesize humans in **novel views**, **novel poses**, and **different styles** with **few samples**. As shown in Table 1 and ablation studies (Section 4.3, Figure 2-3, Table 5-6), all the modules are of great importance to achive state-of-the-art results. We also show that a straighforward combination of some modules would not lead to the desirable results in the ablation studies. It is worth emphasizing that prior works can only achieve **one** or **two** effects (Table 1). We hope that reviewers appreciate more on the improtance of advancing the state-of-the-art with the proposed method (algorithm and implementation).
>
> Our first novelty is that our 4D human video stylization method is the first one that is able to stylize the whole scene **under novel views and poses**, which has not been explored in the existing methods.
>
> Our second novelty is that the proposed geometry-guided triplane can encode features more effective than the vanilla tri-plane. The vanilla tri-plane is sub-optimal when given **sparse views** as the input and will generate blurred prediction and inconsistency among frames. To solve this problem, we encode coordinates by U-Nets to generate more smooth details. Visual comparisons of both tri-planes in Figure 2 and supplementary video [04:29-04:47] shows that our method not only yields better consistency but also superior visual quality. This is further verified by the quantitative results in Table 6, which demonstrates the better consistency of our method.

---

### Official Review · Reviewer_uFnH · 2023-10-31

**Soundness:** 2 fair
**Presentation:** 3 good
**Contribution:** 3 good
**Rating:** 6
**Confidence:** 2

**Summary:**

The paper proposes a novel method for human video stylization that leverages 2 NeRFs to represent backgrounds and the human body separately.
It introduces a geometry-guided tri-plane representation to learn the 3D scene more efficiently and effectively. The proposed method can accommodate novel poses and viewpoints, making it a versatile tool for creative human video stylization.

**Strengths:**

The paper proposes a novel method for human video stylization that leverages 2 NeRFs to represent backgrounds and the human body separately.
The proposed geometry-guided tri-plane representation enhances feature representation robustness compared to direct tri-plane optimization by introducing a geometric prior on the tri-plane. This geometric prior is achieved by discretizing the 3D space as a volume and dividing it into small voxels, with sizes of 10 mm × 10 mm × 10 mm. Voxel coordinates transformed by the positional encoding are mapped onto three planes to serve as the input to the tri-plane.

**Weaknesses:**

The proposed contributions are trivial besides the two Nerf ideas.
The AdaAttN, loss functions are all borrowed from existing methods.
The demo results didn't show large angles of novel views from the backgrounds.

**Questions:**

Have you thought of using DeamFusion-like models to generalize on the background scene generation? Maybe this can help with adding information to your 3D scene.

---

> ### Author Response · Authors · 2023-11-23
>
> **[Q1] The proposed contributions are trivial besides the two Nerf ideas. The AdaAttN, loss functions are all borrowed from existing methods. The demo results didn't show large angles of novel views from the backgrounds.**
>
> **1. Novelty.** We note that while some components are known in the field, it requires **meticulous algorithmic design** to develop an efficient and effective model to synthesize humans in **novel views**, **novel poses**, and **different styles** with **few samples (sparse background views)**. As shown in Table 1 and demonstrated in the ablation studies (Section 4.3, Figure 2-3, Table 5-6), all the modules in the proposed method are of great importance to achive state-of-the-art results, accomplishing all tasks (novel view, animation, stylization, and efficiency). We also show that a straighforward combination of some modules would not lead to the desirable results in the ablation studies. It is worth emphasizing that prior works can only achieve **one** or **two** effects (Table 1). We hope that reviewers appreciate more on the improtance of advancing the state-of-the-art with the proposed method (algorithm and implementation).
>
> Our first novelty is that our 4D human video stylization method is the first method that is able to stylize the whole scene (human and background) **under novel views and poses**, which has not been explored in the existing methods.
>
> Our second novelty is that the proposed geometry-guided triplane is able to encode the features more **effective** than the vanilla tri-plane [1]. The vanilla tri-plane has been widely explored for its **efficiency** compared to the MLPs in NeRFs. However, the existing tri-plane based methods such as [1] optimize the features of three planes directly, which is sub-optimal when given **sparse views** as the input and will generate noisy prediction and inconsistency among frames. To solve this problem, we propose the geometry-guided tri-plane that takes the voxel coordinates transformed by the positional encoding as the input. Three U-Nets are utilized to encode the triplane features, which utilize nearby pixels on the planes to represent the scenes and can generate more smooth details compared to the existing tri-plane structure. The visual comparisons of the proposed tri-plane and the existing tri-plane [1] on the **reconstructed background** is shown in Figure 2. It can be seen that the proposed tri-plane can recover more clear background texture (1st row) and sharper contours (2nd row).  For a more comprehensive set of visual evidence pertaining to reconstruction and stylization, please refer to the supplementary video [04:29-04:47]. Our method not only yields better consistency but also superior visual quality. This is further verified by the quantitative results presented in Table 6, which demonstrates the better consistency of our approach.
>
> [1] Fridovich-Keil et. al. K-planes: Explicit radiance fields in space, time, and appearance, CVPR2023.
>
> **2. Large angles of novel views.** We present the visual examples from large angles of novel views of the proposed method with other state-of-the-art results in the supplementary video [02:29-02:49]. It can be seen that the proposed method strikes a superior balance between stylized textures and temporal coherence.
>
> **[Q2] Have you thought of using DeamFusion-like models to generalize on the background scene generation? Maybe this can help with adding information to your 3D scene.**
>
> Exploring DreamFusion presents a fascinating avenue for future research. We plan to leverage the robust generative capabilities of stable diffusion models to help train our NeRF framework given the text prompts.

---

### Official Review · Reviewer_xdnP · 2023-11-01

**Soundness:** 3 good
**Presentation:** 3 good
**Contribution:** 2 fair
**Rating:** 6
**Confidence:** 4

**Summary:**

This paper proposes editing videos using the 4D NeRF representation. The video is firstly reconstructed as the static background NeRF plus a Neural Avatar. For style transfer, it uses the feature from the NeRF backbone and projects it to the stylized space.  With this canonical representation, it can generate consistent results when editing viewpoint, human pose, and style.

**Strengths:**

The overall idea is technically sound. Using NeRF-like canonical representation can solve video editing problems in a unified framework.

**Weaknesses:**

(1) Limited technical novelty. The NeRF reconstruction part is almost identical to NeuMan except that it uses a tri-plane representation, which has also been exploited widely. This choice is straightforward. The style transfer part follows AdaAttn without any modification. So the main paper is a straightforward combination of two existing works.


(2). The overall visual quality is low, as shown in the paper and supplement material. Editing viewpoint and human pose is not new as this part is almost identical to the NeuMan.  But for stylization, compared to existing baselines, the qualitative experiments are not enough, just less than 5 style transfer results are shown.

(3). The comparison to existing baselines is not fair. This paper stylizes foreground and background separately while baseline methods are applied in a foreground-agnostic manner. I believe it is easy to extend existing work to be segmentation-aware.

(4). Lacking comparison with other video editing baselines which also exploited a layered and canonical representation. e.g.,
Layered Neural Atlases for Consistent Video Editing, SIGGRAPH Asia 2021,
This baseline can be easily extended using AdaAttn by optimizing the canonical atlas.

(5) Minor wriring issues
In Equ.5. \mathcal{E}(x) should also depends on frame indices.

**Questions:**

Please address the weakness (1),(2),(3) and optionally (4)

---

> ### Author Response · Authors · 2023-11-23
>
> **[Q1] Limited technical novelty. The NeRF reconstruction part is almost identical to NeuMan except that it uses a tri-plane representation, which has also been exploited widely. This choice is straightforward. The style transfer part follows AdaAttn without any modification. So the main paper is a straightforward combination of two existing works.**
>
> We note that while some components are known in the field, it requires **meticulous algorithmic design** to develop an efficient and effective model to synthesize humans in **novel views**, **novel poses**, and **different styles** with **few samples (sparse background views)**. As shown in Table 1 and demonstrated in the ablation studies (Section 4.3, Figure 2-3, Table 5-6), all the modules in the proposed method are of great importance to achive state-of-the-art results, accomplishing all tasks (novel view, animation, stylization, and efficiency). We also show that a straighforward combination of some modules would not lead to the desirable results in the ablation studies. It is worth emphasizing that prior works can only achieve **one** or **two** effects (Table 1). We hope that reviewers appreciate more on the improtance of advancing the state-of-the-art with the proposed method (algorithm and implementation).
>
> Our first novelty is that our 4D human video stylization method is the first method that is able to stylize the whole scene (human and background) **under novel views and poses**, which has not been explored in the existing methods.
>
> Our second novelty is that the proposed geometry-guided triplane is able to encode the features more **effective** than the vanilla tri-plane [1]. The vanilla tri-plane has been widely explored for its **efficiency** compared to the MLPs in NeRFs. However, the existing tri-plane based methods such as [1] optimize the features of three planes directly, which is sub-optimal when given **sparse views** as the input and will generate noisy prediction and inconsistency among frames. To solve this problem, we propose the geometry-guided tri-plane that takes the voxel coordinates transformed by the positional encoding as the input. Three U-Nets are utilized to encode the triplane features, which utilize nearby pixels on the planes to represent the scenes and can generate more smooth details compared to the existing tri-plane structure. The visual comparisons of the proposed tri-plane and the existing tri-plane [1] on the **reconstructed background** is shown in Figure 2. It can be seen that the proposed tri-plane can recover more clear background texture (1st row) and sharper contours (2nd row).  For a more comprehensive set of visual evidence pertaining to reconstruction and stylization, please refer to the supplementary video [04:29-04:47]. Our method not only yields better consistency but also superior visual quality. This is further verified by the quantitative results presented in Table 6, which demonstrates the better consistency of our approach.
>
> [1] Fridovich-Keil et. al. K-planes: Explicit radiance fields in space, time, and appearance, CVPR2023.
>
> **[Q2] The overall visual quality is low, as shown in the paper and supplement material. Editing viewpoint and human pose is not new as this part is almost identical to the NeuMan. But for stylization, compared to existing baselines, the qualitative experiments are not enough, just less than 5 style transfer results are shown.**
>
> We present additional visual examples that showcase 9 more styles in the supplementary video on novel views and poses [01:20-03:29]. Compared to the existing methods, our method demonstrates better consistency and visual quality. We note that other state-of-the-art methods could not achieve **all** effects achieved by our model.
>
> **[Q3] The comparison to existing baselines is not fair. This paper stylizes foreground and background separately while baseline methods are applied in a foreground-agnostic manner. I believe it is easy to extend existing work to be segmentation-aware.**
>
> To extend existing works to be segmentation-aware, we integrate Swin Transformer [2] and SAM [3] to segment foreground from the background. Initially, Swin Transformer is utilized to segment the foreground, which generates foreground human with coarse boundaries. Then points are randomly sampled inside the foreground and served as the input of SAM which refines the segmentation by predicting more precise human contours. The visualization can be found in the supplementary video [03:31-04:27]. It can be seen that our method outperforms the existing methods in generating stylization results, maintaining consistent lighting between the foreground and background, and ensuring smoother transitions at the boundaries of the human.
>
> [2] Liu et. al. Swin transformer: Hierarchical vision transformer using shifted windows, ICCV2021.
> [3] Kirillov et al. Segment anything. arXiv preprint arXiv:2304.02643, 2023.

---

### Meta-Review · Area_Chair_cFMc · 2023-12-08

**Metareview:**

This paper introduces a human video stylization method, utilizing the 4D NeRF (Neural Radiance Field) representation to separately model the static background and human body. The technique combines a static background NeRF with a Neural Avatar, employing a tri-plane representation for efficient  3D scene learning. By operating within the NeRF-rendered feature space, the method enables consistent editing of viewpoints, human poses, and styles, allowing for creative and versatile video modifications.

The technical contribution of this work is the integration of existing modules and its application to a bit new task.

The three reviewers (Scores: 6, 6, 5->6->5 (smP6 mentioned the score was upgraded but downgraded later) ) acknowledged the effectiveness of the proposed method and left several comments.
Their key concerns include the limited visual quality of the results, lack of technical novelty, and lack of comprehensive comparative analysis (weak and unfair baselines).

The AC read all the reviews and rebuttals. After the rebuttal, the comparison issues are a bit mitigated. Also, Reviewer xdnP acknowledged the improvement in the quality by the rebuttal materials.

However, the technical novelty concern still remains across all the reviewers as can be seen in
each reviewer's final comments:

- Reviewer xdnP raised the score but didn't champion this work by the novelty concern.

- Reviewer smP6 mentioned "partly addressed the concerns."

- Reviewer uFnH didn't reply to the rebuttal, but the novelty concern was the main concern of the reviewer which still remains.

Besides, in the rebuttal, the authors mentioned that "Our first novelty is that our 4D human video stylization method is the first one that is able to stylize the whole scene under novel views and poses, which has not been explored in the existing methods."
However, this AC found that there is a missing citation [C1], where they also presented 4D human video stylization, which should have been compared.

[C1] T. Nguyen-Phuoc et al., "SNeRF: Stylized Neural Implicit Representations for 3D Scenes," Trans. on Graphics, 2022.

Given the concern agreed across all the reviewers, the AC concludes that the novelty of the work remains marginal.

**Justification For Why Not Higher Score:**

While the two reviewers rate the score favorably, the comments and concerns actually do not fully support the acceptance of this work. The reviewers mentioned that the rebuttal "partially" addressed the concerns. This AC concurs with the reviewers' concern about the technical novelty shared across all the reviewers despite the rebuttal response and missing a closely related citation.

**Justification For Why Not Lower Score:**

.N/A

---

### Decision · Program_Chairs · 2024-01-16

Reject